# Diabetes and Cardiovascular Diseases Risk Assessment in Community Pharmacies: An Implementation Study

**DOI:** 10.3390/ijerph19148699

**Published:** 2022-07-17

**Authors:** Sarah Rondeaux, Tessa Braeckman, Mieke Beckwé, Natacha Biset, Joris Maesschalck, Nathalie Duquet, Isabelle De Wulf, Dirk Devroey, Carine De Vriese

**Affiliations:** 1Department of Pharmacotherapy and Pharmaceutics, Faculty of Pharmacy, Université Libre de Bruxelles (ULB), 1050 Brussels, Belgium; sarah.rondeaux@ulb.be (S.R.); natacha.biset@ulb.be (N.B.); 2Department of Family Medicine and Chronic Care, Faculty of Medicine, Vrije Universiteit Brussels (VUB), 1090 Brussels, Belgium; tessa.braeckman@vub.be (T.B.); mieke.beckwe@vub.be (M.B.); dirk.devroey@vub.be (D.D.); 3Association of Pharmacists Belgium, 1000 Brussels, Belgium; joris.maesschalck@apb.be (J.M.); nathalie.duquet@apb.be (N.D.); isabelle.dewulf@apb.be (I.D.W.)

**Keywords:** community pharmacy services, implementation, risk assessment, diagnostic screening programmes, diabetes mellitus, type 2/prevention and control, cardiovascular diseases/diagnosis/epidemiology/prevention and control, Belgium

## Abstract

The implementation of a new service is often challenging when translating research findings into routine clinical practices. This paper presents the results of the implementation study of a pilot project for a diabetes and cardiovascular diseases risk-assessment service in Belgian community pharmacies. To evaluate the implementation of the service, a mixed method was used that follows the RE-AIM framework. During the testing stage, 37 pharmacies participated, including five that dropped out due to a lack of time or COVID-19-related temporary obligations. Overall, 502 patients participated, of which 376 (74.9%) were eligible for according-to-protocol analysis. Of these, 80 patients (21.3%) were identified as being at high risk for the targeted diseases, and 100 (26.6%) were referred to general practice for further investigation. We presented the limited effectiveness and the key elements influencing optimal implementation. Additional strategies, such as interprofessional workshops, a data-sharing platform, and communication campaigns, should be considered to spread awareness of the new role of pharmacists. Such strategies could also promote collaboration with general practitioners to ensure the follow-up of patients at high risk. Overall, this service was considered easy to perform and feasible in practice but would require financial and external support to ensure its effectiveness, sustainability, and larger-scale implementation.

## 1. Introduction

Noncommunicable diseases (NCDs) such as cardiovascular diseases and diabetes are slow-progressing and long-duration diseases that result from a combination of genetic, environmental, physiological, and behavioural factors. Their socioeconomic burden is recognised universally, and they have a large impact on public health and on quality of life [1]. To lessen the impact of diabetes and cardiovascular diseases and their complications, individual and collective approaches should be implemented that focus on reducing the modifiable risks associated with these diseases. In addition, cost-efficient interventions to prevent and control the diseases should be implemented. Preventive measures such as early detection and intervention have shown to be effective economic investments. They delay progression and the onset of complications and may reduce the need for expensive treatment if provided early [2,3]. Healthcare practitioners play a key role in tackling the burden of NCDs. However, the increased burden on healthcare providers of managing population health and related costs obliges health systems to consider new strategies in order to use their resources more efficiently. Such strategies include delivering preventive services. As such, community pharmacists are ideally placed. They are often the first point of contact between patients and the healthcare system due to their convenient locations, extended opening hours, and the provision of pharmaceutical care without an appointment [4].

Recently, the pharmacist’s role has shifted slowly from the traditional one of medication dispensing to being a provider of services and information, particularly regarding the better use of medicines [5,6]. Consequently, pharmacist-led screening models have been increasingly researched due to their potential benefit to public health [7]. Several studies have demonstrated that community pharmacy screening is feasible and effective at revealing unknown cases of diabetes and cardiovascular diseases [8,9].

The Brussels-Capital Region has not escaped the dangerous rising prevalence of NCDs. This is due to its ageing population, migration-induced diversity, and social inequalities [10,11]. Moreover, the Brussels context faces another challenge—a significant shortage of general practitioners who would usually provide screening services [12]. Patients face difficulties in booking appointments and often visit the nearest hospital emergency department directly for treatments. This disrupts the usual health screening processes and, so, case detection. In Belgium, pharmacists are still principally medication dispensers, despite pharmaceutical associations advocating for an extension of their role. Together with the knowledge that early detection through screening in community pharmacies is known to be valuable and achievable, a few short-term pilot studies have been carried out in a limited number of pharmacies in Belgium to assess the uptake of diabetes screening. However, none have been implemented at a larger scale or evaluated the implementation process [13,14].

Although international evidence shows the benefit of pharmacy-led screening services, translating the findings into practice can be challenging due to the variability of regulatory environments and practices between countries. Implementing new, national pharmaceutical services is a complex process, and high-quality service provision is not simple to achieve [15,16]. There is a difference between efficacy (outcome of an intervention under ideal conditions) and effectiveness (outcome of an intervention under normal conditions) [17]. As such, when translating research into practice in the real world, it is essential to evaluate more than effectiveness alone to understand the means and conditions under which the intervention worked or did not work. This approach also allows researchers to gain a deeper understanding of the feasibility and sustainability of interventions by highlighting facilitators and barriers that were not foreseen prior to the implementation. To help to bridge implementation gaps, implementation studies of new pharmaceutical services are gradually emerging in the literature. These have become recognised as critical to ensuring service quality and, therefore, their effectiveness [18,19].

Given the high density of pharmacies in the Brussels-Capital Region and the challenges faced due to the local context, the Association of Belgian Pharmacies (APB) planned to conduct an implementation pilot project among a larger sample of pharmacies. Research teams from two universities collaborated to evaluate the feasibility of offering diabetes and cardiovascular disease screening services in community pharmacies. An implementation study was conducted to identify the contextual factors facilitating or hindering a successful implementation within community pharmacies and to explore the experiences of pharmacists and patients. This paper aims to present the results of the implementation study conducted around a screening service.

## 2. Materials and Methods

### 2.1. Description of the Screening Service

From October 2020 to February 2021, a selection of Brussels pharmacists invited patients to participate in opportunistic screening for diabetes and cardiovascular disease risk factors. They specified that this new service was part of a qualitative and quantitative research study to evaluate its feasibility in community pharmacies. The screening followed a multi-step method illustrated in Figure 1. First, the pharmacist assessed the patient’s eligibility based on the inclusion and exclusion criteria. The screening then evaluated the diabetes risk in the patient, followed by an assessment for cardiovascular diseases. Finally, the need to consult a general practitioner or specialist was determined based on the outcome of the risk assessment.

All Brussels residents between 40 and 65 years old were eligible. The screening was free of charge in order to be accessible to the most vulnerable and socially disadvantaged individuals. Considering the high risks that correlate with certain ethnic backgrounds, the inclusion criteria were widened to 25-years-old individuals from North-African, South-Asian, and Turkish backgrounds, as suggested by the guidelines for opportunistic screening for diabetes in Belgium [24,25]. To avoid including patients already diagnosed with the targeted diseases, patients were excluded if they were followed regularly by a general practitioner or taking medication for diabetes, cardiovascular diseases, or kidney diseases.

The first evaluation was based on a translation of the Finnish Diabetes Risk Score (FINDRISC) questionnaire, a validated questionnaire that aims to identify individuals at high risk of developing type 2 diabetes mellitus within 10 years [20]. The questionnaire is a reliable, valuable, and easy-to-use screening tool recommended by medical guidelines for diabetes and prediabetes screening in Belgium [26,27]. It contains eight questions regarding the patient’s demographic characteristics, medical history and lifestyle behaviour. Individuals with a score > 7 were invited to have their haemoglobin A1C (HbA1c) measured with a point-of-care testing (POCT) device to refine their risk profile further.

The cardiovascular evaluation was based on the Boland algorithm [21]. This evaluation relied on medical history and blood pressure measurements. The following six risk factors were considered: age [A], blood pressure [B], cigarette smoking [C], type 2 diabetes mellitus [D], a personal ischaemic event [E], and a familial ischaemic event [F]. As patients with a history of diabetes or cardiovascular diseases were not eligible, only the other four factors were assessed (ABCF). If the patient presented one of the four risk factors and did not present the smoking-related risk exclusively, the evaluation of their cardiovascular risk was based on the calibrated SCORE chart (Systematic Coronary Risk Evaluation) for Belgium. This estimation tool is based on age, gender, total cholesterol, and smoking behaviour and establishes the 10-year fatal risk of cardiovascular diseases. It was demonstrated to be suitable, accurate, and precise for the assessment of cardiovascular risk in Belgium [22]. Since the cholesterol value was not measured during this project, the mean of the Belgian cholesterol value for men and women was used to evaluate the score of the patients. Individuals were considered at low to moderate cardiovascular risk if they either had no risk factors (ABCF-), if the cardiovascular risk was only associated with smoking (C+), or if the result of the SCORE table was <5%. Individuals were considered at an increased cardiovascular risk if they had a SCORE between 5 and 9% and were considered at high cardiovascular risk if they had a SCORE ≥ 10%.

The pharmacist provided counselling and personalised advice to patients with low and moderate risk profiles to help them manage the risk factors detected. However, patients presenting a combination of a FINDRISC score between 7 and 11 and an HbA1c reading of ≥6.5%, patients with a FINDRISC score ≥ 12 and an HbA1c reading ≥ 5.7%, patients with a cardiovascular SCORE ≥ 10, and patients with a cardiovascular SCORE between 5 and 9 presenting high blood pressure values or a waistline of ≥80 cm for women and ≥90 cm for men, were considered to be at high risk and oriented to a general practitioner for further investigation (Figure 1).

### 2.2. The Organisation of the Screening Service

Before implementing the screening project within their pharmacies, pharmacists followed a training session on the risk-determination procedures. This session included the project protocol, the use of the POCT device and how to perform an HbA1c measurement correctly, the hygienic measures to be taken considering the COVID-19 epidemic and motivational techniques for patient counselling. The participating pharmacies received a set of promotional materials to increase the visibility of the project and educational materials to initiate and support a dialogue between the pharmacist and patients. The POCT device was installed within the pharmacies, and the reliability of the results was monitored by City-Labs, the medical laboratory of the Cliniques universitaires Saint-Luc. Additionally, the APB designated a target of 30 patients per pharmacy to boost pharmacies’ involvement and motivation within the project.

### 2.3. Study Design

To evaluate the implementation of the screening service, a prospective and observational study using a mixed method of quantitative and qualitative analysis was conducted following the RE-AIM model [28]. This implementation framework is one of the most widely used frameworks for planning and evaluating public health and medical interventions [29]. It has five dimensions: reach (R), effectiveness (E), adoption (A), implementation (I), and maintenance (M). The collected outcomes were defined to characterise the RE-AIM dimensions as closely as possible and explore the potential barriers and facilitators of these dimensions while being adapted to the context and objectives of a pilot study. The collected outcomes are described in Table 1.

### 2.4. Participants

Patients willing to participate in the screening were asked if they consented to their data being used for scientific analysis and/or if they consented to be contacted by the university research team for a short interview on their participation. All of the pharmacists who participated in the project were eligible for the implementation study.

### 2.5. Data Collection

The quantitative data were collected through a web platform designed for this pilot project, in which pharmacists recorded patient information during the screening. A data file containing patients’ demographic characteristics (age, gender), responses to the questionnaires, and their risk determination results (FINDRISC category, HbA1c results, and cardiovascular risk category) was later shared with the research team for analysis. The data file also contained contact information for those patients who consented to be contacted for a follow-up interview.

The qualitative data were collected during semi-structured individual interviews with patients and semi-structured focus groups with participating pharmacists. The interview guides, one for the patients and one for the pharmacists, were based on the implementation outcomes, which followed the RE-AIM framework.

One month after inclusion, patients were randomly selected and invited by one of the three researchers to participate in a telephone interview. The semi-structured interviews were conducted in French or Dutch, according to the patient’s preference. After each semi-structured interview, the interviewer completed a data saturation file to determine whether new information had been collected or not. Data saturation was discussed weekly and considered to be reached when no new information had been collected during three consecutive interviews.

At the end of the project, a researcher contacted the participating pharmacists to invite them to participate in focus groups. If a pharmacist was willing to participate but unavailable on the dates proposed, an individual interview following the focus group interview guide was suggested to ensure data saturation. Focus groups and interviews were organised and conducted by two researchers virtually through the Zoom platform. Pharmacists who did not include any patients in the project were also contacted by telephone and interviewed to understand the reasons for non-inclusion. Possible saturation was discussed between the researchers and considered reached when no new information had been collected during the last focus group.

### 2.6. Data Analysis

All of the interviews were audio-recorded and transcribed ad verbatim. Pharmacists’ and patients’ transcriptions were coded separately. The first three transcriptions were coded inductively by two independent researchers using the Nvivo software. The codes were compared and matched, and discrepancies were discussed. The codes were grouped into themes and classified according to the dimensions explored. The following transcriptions were coded using the pre-established coding tree. Every three interviews, coding discrepancies were discussed, and new codes were added to the coding tree. The quantitative data of patients who consented to data analysis were analysed descriptively with Excel.

### 2.7. Ethics Approval

The study protocol was submitted to the Saint-Luc-UCL (Belgium) ethics committee in May 2020 and received ethics approval in July 2020 (B40321836258).

## 3. Results

The screening service was organised consecutively in two overlapping phases, from 1 October 2020 to 26 January 2021 and from 5 January 2021 to 31 March 2021. The phases were organised with 24 pharmacies and 18 pharmacies, respectively, including five pharmacies that participated in both phases. To evaluate the service implementation, a total of 20 semi-structured interviews were conducted with patients who participated in the screening programme.

Four focus groups with a total of 13 pharmacists were organised via the Zoom platform. In addition, three individual interviews were conducted by telephone with pharmacists that could not attend the focus groups.

Several key themes that influenced the implementation of the screening service were identified during this qualitative research. The themes, with illustrative quotes and classified following the RE-AIM framework, are presented in Appendix A for patients and Appendix B for pharmacists.

### 3.1. Reach

During the study, a total of 502 patients were enrolled in the screening programme by 32 pharmacies. On average, every pharmacy included 16 patients, with a minimum enrolment of one patient and a maximum of 41 patients.

#### 3.1.1. Description of the Patient Characteristics

Of the 502 patients, 411 (81.9%) consented to the analysis of their data for scientific purposes, including patients older than 65 years (8.5%). Excluding this latter group resulted in 376 patients eligible for the according-to-protocol analysis. Most patients included were female (61%) and aged between 25 and 65 years, with a median of 50 years old.

#### 3.1.2. Proportions of Risk Profiles

The 329 patients with a FINDRISC score of ≥7 (87.5%) were invited to have their Hb1Ac level measured. The HbA1c value varied from 2.9% to 6.3%, with a median of 5.4%. This measurement was not performed for five patients (1.3%), which resulted in an undetermined risk profile. The descriptive results are presented in Table 2. Of all patients, 145 (38.6%) and 152 (40.4%) with low and moderate risk profiles, respectively, received counselling on managing their determined risk factors. However, 74 patients (19.7%) were established to be at a high risk of developing diabetes and were referred to a general practitioner for further evaluation.

Of the 297 patients with a low or moderate risk profile for diabetes, 255 (85.9%) participated in the evaluation of cardiovascular diseases. Of these, 67 (26.3%) were assessed to be at a low cardiovascular-diseases risk as they presented no clinical risk factors [ABCF−] or only cigarette smoking [C+]. The risk was further assessed using the calibrated SCORE chart for the 188 patients (71.8%) presenting any other risk factors [ABF+]. The distribution of the risk profile based on the absence/presence of the medical risk factors and associated SCORE results is presented in Table 3. Overall, 223 patients (89.6%) were assessed to have a low/moderate risk profile and received explanations on cardiovascular-diseases risk management. However, six patients with a score of ≥10 (2.4%) and 20 patients (8.0%) with a SCORE between 5 and 9 presenting high blood-pressure values or a waistline of ≥80 cm for women and ≥90 cm for men were assessed to be at a higher risk and were referred to a general practitioner.

#### 3.1.3. Reasons for Participating in or Rejecting the Risk Assessment

The general perception of the usefulness of the availability of screening services was encouraging across the board. Patients felt that the threshold for visiting a pharmacist is much lower than for making an appointment with a general practitioner. Some patients were driven by curiosity, while others realised they had not seen a general practitioner for a long time and perceived it as an opportunity to learn more about their health. Other motivators were the fear of the detrimental effect of the targeted diseases on their health, having seen the consequences and sequelae of diabetes and cardiovascular disease among family members. Lastly, trust in the pharmacists contributed to the patients’ positive attitude towards this preventative intervention.

On the other hand, pharmacists shared that a few patients refused to participate due to their fear of what the outcome might be. These patients preferred to turn a blind eye to risks and declined. For others, the lack of time or availability to take part in the screening was shared as the main reason for not participating.

#### 3.1.4. Barriers and Facilitators to Service Proposal and Patient Participation

Among the pharmacists, some used an active approach in which the pharmacist presented the risk assessment as an opportunity for patients to learn more about their risks and general health. The emphasis on the lack of charge for the service and partnership with universities also seemed to influence positively the patients’ willingness to participate. Although some pharmacists had no difficulty in proposing the programme directly to patients, others found this step to be the most delicate. Despite reporting few refusals, some pharmacists feared that the patients they did not know well might feel stigmatised by the proposal. They also often worried that patients would feel forced to participate and thus refrained from proposing the risk evaluation actively to eligible patients. Other barriers, such as a language barrier or lack of time, sometimes held pharmacists back from asking patients to participate. Aside from the active approach, awareness of the project was indirectly facilitated by the visual promotional material, which generated curiosity among some patients. As an alternative to distributing flyers, a few pharmacists preferred to hand out the FINDRISC questionnaire. This allowed patients to fill in the questionnaire at home and then come back to continue the risk-assessment procedure at their convenience.

Finally, some pharmacists expressed frustration with the age limit criteria, which hindered some potential inclusion. In their opinion, broadening the inclusion criteria could only be beneficial as it could be an opportunity to deliver a preventive message and raise awareness of diabetes and cardiovascular diseases.

### 3.2. Effectiveness

#### 3.2.1. Outcome of Medical Follow-Up of Patients Identified as High-Risk

The effectiveness study was evaluated exclusively during the first phase of the project. During this period, 51 patients (23.2%) out of the 220 screened were assessed as having a high risk of developing either diabetes or cardiovascular diseases. Eighteen patients did not consent to be contacted, leaving 33 possible inclusions for the effectiveness analysis. In the end, only 20 patients could be contacted.

Two patients did not seem to be aware of their high-risk profile, declaring it was not communicated by the pharmacist. Some patients prioritised other medical problems or postponed taking a medical appointment until there was a decrease in the incidence of COVID-19-positive cases. Six patients were already in medical follow-up for either diabetes or cardiovascular diseases. Only six patients received further analysis after the risk assessment, with five not diagnosed with diabetes or cardiovascular diseases and one diagnosed with diabetes. The results are presented in Table 4.

#### 3.2.2. Patient Perception of and Attitude during the Risk Assessment

The patients were at ease during the procedure, although pharmacists reported that most patients did not feel comfortable pricking their fingers themselves and thus always asked the pharmacists for assistance. Those patients for whom an HbA1c measurement was unnecessary thought they would be more reassured of the validity of the low risk if they could see the result from the HbA1c. The communication by the pharmacists was highly appreciated by the patients and considered clear, reassuring, and respectful. The patients who received a printout of the results did not think it was necessary to receive it. On the other hand, when there was no printout available, this was perceived as a shortcoming as the lack of tangible information could lead patients to forget the purpose and outcome of the screening.

### 3.3. Adoption

A total of 37 pharmacies participated. Five were considered to have dropped out (13.5%) as they did not include any patients in the study process.

#### Pharmacists’ Reasons for Adopting or Rejecting the Service

Most participating pharmacists expressed that they saw the study as an opportunity to show their added value. In addition, most of them perceived that services related to prevention practices and screening would be part of the future of their profession. Most of the pharmacists who dropped out during the project reported that the COVID-19 crisis was the main reason. Some lacked the workforce or time to devote to the pilot project, while others felt uncomfortable taking the patients into a restricted pharmacy area at a time when they knew that sanitary conditions were stricter and contamination heightened.

### 3.4. Implementation

#### 3.4.1. Internal Organisation and Adaptation to Implement the Project

Considering the restrictions due to the sanitary crisis, only one pharmacist per pharmacy could attend the training. Depending on the pharmacy, some pharmacists shared that they trained their team afterwards so everyone within their pharmacies could perform the service. However, some lacked the confidence to deliver the service.

When starting the project, the community pharmacists first preferred to book appointments with the patients. The pharmacists noticed that in some cases, the patients did not attend on the specific date, either because they had forgotten or because their interest had waned over time. Therefore, once pharmacists felt more comfortable and experienced with the procedure, they tried to perform the screening directly, without appointments, which led to a better uptake.

#### 3.4.2. Time Needed to Provide the Service

The reported duration of performing the service varied among the pharmacists, with an average of 15 to 30 min for most patients. However, the screening sometimes revealed that some patients needed to talk and raised various questions, leading to further counselling that sometimes lasted up to an hour. Nevertheless, the pharmacists agreed that the risk evaluation should be maintained between 20 and 30 min for the service to be sustainable.

#### 3.4.3. Facilitators and Barriers for Optimal Implementation

To carry out the project, pharmacists expressed that having a dedicated screening space out of sight of other patients was valuable for making the patients feel at ease. Dedicating a confidential area for prevention services would therefore be helpful in the future for pharmacies that currently lack an available space. Additionally, pharmacists shared that it was necessary to have enough staff to offer this prevention service so that the rest of the team could remain available to provide pharmaceutical care at the counter and perform other routine tasks. Similarly, the lack of time or workforce appeared to be a recurring hindrance to some pharmacists, preventing them from screening patients on the spot and obliging them to book appointments at a later, more convenient time.

#### 3.4.4. Adherence to the Protocol

Among the 411 patients who consented to the analysis of their data, 35 (8.5%) were older than 65 years despite the protocol imposing their exclusion. According to most pharmacists, the age range was considered too restrictive because some patients over 65 were not necessarily regularly followed by a general practitioner and could have benefited from the risk evaluation. Thus, some pharmacists explained that although they were obliged to refuse spontaneous requests from such patients, they did not feel comfortable doing so.

### 3.5. Maintenance Outlook

#### 3.5.1. Pharmacist Satisfaction and Service Acceptability to Patients

All patients felt they received sufficient information and that the pharmacists took the time to make them feel comfortable. Overall, the risk assessment allowed the pharmacists to spend time with the patients, which was appreciated and helped to initiate a personal dialogue and deepen the pharmacist–patient relationship. The pharmacists were eager to continue the service, expressing a desire to repeat the pilot study if the opportunity arose.

#### 3.5.2. Sustainability Factors

The pharmacists mentioned that they would like to increase their collaboration with other professionals, especially general practitioners, to maintain this type of preventative service. They felt that this would strengthen the follow-up of the patients. It was suggested that it could be put into practice through medico-pharmaceutical meetings to develop a collaborative approach to the services. To enable this collaboration, patient outcomes and measurement data could be shared through an electronic health platform. This should be conducted with the patients’ consent and preferably through an established health pathway. Most pharmacists felt that the training preceding the implementation was sufficient. However, it was suggested that training in preventive-related services should be integrated into the pharmacy master’s curriculum if these services became systematic.

The patients and pharmacists expressed their view that the risk evaluation should remain free of charge for the patient. On the other hand, the financial cost for pharmacists was discussed. Some pharmacists expressed the need that remuneration was essential to compensate for the time invested. Others talked about receiving financial compensation for the costs of the POCT device and consumables. To avoid large expenses, a pharmacist suggested that the POCT machine could be rotated monthly between groups of pharmacies that would offer the prevention service for a short period each year. This proposal was appreciated as it would also allow pharmacists to prepare for the service in a similar way as for a health campaign and to recruit new patient participants in advance.

Finally, patients and pharmacists remarked that the awareness campaign needed a broader approach since the pharmacist had to invest a lot of time in approaching possible at-risk patients. Campaigns through the community were mentioned as a possibility, while others suggested additional media coverage.

## 4. Discussion

This research is the first study in Belgium examining patients’ and pharmacists’ perceptions of a risk-assessment service for diabetes and cardiovascular diseases and the factors influencing its implementation in community pharmacies. The positive attitude of the patients toward the risk assessment and the overall enthusiasm of the participating pharmacists are encouraging factors for the sustainability of the service. The interviewed pharmacists considered the risk assessment to be an implementable service in practice. However, adaptations at the pharmacy level and within the healthcare system are necessary to ensure its effectiveness and enable a successful long-term and large-scale implementation.

The acceptability of the service to patients was directly linked to the perception that prevention is undervalued in the current healthcare system and that the high accessibility of community pharmacies could help more patients to be reached who are unaware of their risk. However, pharmacists reported that the main difficulty was proposing the service to the patients due to the fear of stigmatisation and potential negative patient reactions. While some pharmacists found indirect approaches to recruiting patients through promotional and other eye-catching materials, some were reluctant to propose the service despite the limited number of refusals. This fear of offering the risk assessment seems to be correlated to the novelty of pharmacy-based screenings in Belgium and the hesitation to overstep their role in the eyes of the patients, which is typically perceived as limited to medication provision and information [30]. Although a review confirms that individuals participating in pharmacy-based screening were consistently satisfied [31], those initiatives are still in an early stage of implementation, and evidence suggests that a better understanding of the pharmacist’s services could increase patient uptake [6]. Therefore, increasing public awareness of this new service through community approaches and additional media coverage would help acknowledge and reinforce the perception of this broader role of pharmacists.

In this study, the pharmacists’ enthusiasm for prevention-related services was directly correlated to their perception of a shift in their practice beyond the traditional dispensing role. This perceived shift makes pharmacists’ work more relevant and thus more appealing, echoing the findings of previous research [32,33]. Considering the widespread enthusiasm for implementing professional services in literature, Siu et al. suggested that implementation discrepancies such as local external factors, individual capabilities, and organisational capacity might be more important sources of variation rather than motivation alone. These discrepancies would thus lead to differences in the uptake of the services [34]. As such, the appeal of participating in pharmacy-led screening services is mainly influenced by the population demographics and the ease of timely access to a general practitioner [35]. To reach the individuals that would benefit the most from the screening service, patients and pharmacists located in areas with a lower social-economic profile stated that it was essential to offer the risk assessment free of charge. They expressed the view that this would maintain high accessibility for potentially high-risk patients who are more likely to face multiple barriers to accessing primary care [36]. Another way to facilitate the uptake was the possibility of performing the risk assessment on-demand, directly when the patient enters the pharmacy, rather than through booking an appointment. However, this capacity to deliver the service is not always feasible in practice and directly correlates to the availability of time and staff, which were highlighted as the main barriers to successful service implementation by previous research [37]. One suggested strategy to overcome this barrier was to hire another pharmacist to cover the time used to perform the risk assessment. However, this would be associated with an extra economic load.

Generally, to sustain services, pharmacies have to adjust the service sufficiently to accommodate for funding changes in order to maintain financial profitability and/or experience the non-financial advantages of the service [18]. For the latter, the pharmacists perceived the service as an opportunity to create a solid personal relationship with the patient, which appears to have more influence on patient loyalty than the technical quality of the provided service [38]. However, to overcome the financial burden, the pharmacists proposed different strategies, including the possibility of having the service approved on a political level. Remuneration reflecting the time spent providing the service and the significant long-term expenses related to the POCT and consumables should be considered. In a cost-effectiveness study for a similar diabetes screening protocol, Wright et al. estimated the cost per person screened as being GBP 28.65 [39]. In our pilot study, the pharmacists were remunerated at a rate of EUR 30.00 per patient, which seemed appropriate for the average service delivery time (~20 min). Alternatively, a suggested strategy received positively by the pharmacists was a system of rotating the POCT equipment monthly between a sample of pharmacies. This organisation would allow pharmacists to promote the risk assessment ahead of time. It would also allow them to make the most of their patient pool available for recruitment, which tends to decrease over time, and thus would potentially increase efficiency over the service delivery period.

The pharmacists reported that the operationalisation of the service turned out to be straightforward without significant complications. This suggests that the different tools and training were sufficient for pharmacists’ self-efficacy, which was demonstrated as a key determinant for motivating community pharmacies to deliver prevention services [32]. However, the restrictions imposed by the COVID-19 crisis limited the training to a single pharmacist per pharmacy, requiring pharmacists to train their staff independently. This approach was sometimes insufficient to acquire the required level of self-efficacy to provide the service and, thus, limited staff adoption of the service. As such, additional training and e-learning could be developed to support other staff members to adopt the service in the future.

The evaluation of the effectiveness of the present study highlighted some weaknesses. However, those findings need to be considered with caution due to the small sample of referred interviewed patients. The lack of awareness of some patients of their high risk following the risk assessment might be explained by the absence of a printout of the results for some or by the pharmacist’s reassuring tone. However, the lack of understanding of a high-risk status could lead to poor attendance in general practice. A review and recent research demonstrated that many referred high-risk patients do not attend medical follow-ups [8,40]. Another limitation was the poor adherence to the eligibility criteria, with the inclusion of patients with an already determined risk or in medical follow-up. Determination of eligibility based on self-reporting by the patients was shown to be less reliable since patients can be unaware of their medical antecedents. Developing a check algorithm to assess and verify eligibility criteria within the web tool may avoid screening non-eligible patients. Additionally, an electronic health record system that would allow pharmacists to share patient data with their general practitioner was suggested for patient-care continuity. Establishing clinical community links between providers and increasing the use of electronic health records to manage patients have been demonstrated to be facilitators for referral [41]. Furthermore, enhancing interprofessional collaboration is crucial to strengthening the effective functioning of services in primary care. In this study, pharmacists suggested that an integrated patient pathway, focusing on a collaborative approach, would be crucial for maintaining the service. Interprofessional education and workshops connecting pharmacists and physicians can be beneficial to increasing awareness and understanding of their respective roles in delivering patient care, which would optimise long-term interprofessional collaboration [42].

Finally, legalising the provision of pharmacy services by the Minister of Health or other regulatory bodies is a determinant of their sustainability and larger-scale implementation [43]. In Belgium, the law requires a POCT policy under the coordination and supervision of a clinical laboratory. However, there is no legal framework regarding the use of POCT outside of a hospital, even though, when combined with a risk-assessment questionnaire, they may provide more accurate screening and referral. To overcome this lack of regulation, a proposal for POCT to be reimbursed was submitted under the condition that the tests would be carried out within an extended legal framework [44]. The elimination of regulatory barriers is an important step toward ensuring optimal patient care and the sustainability of pharmacist-led screening.

In this pilot study, professional associations and academic researchers worked hand in hand to evaluate the feasibility of a risk-assessment service in community pharmacies. Evaluating the implementation of a programme enables the identification of facilitators and barriers according to a location-specific practice. This pilot project included both French- and Dutch-speaking pharmacists from the Brussels-Capital Region. As such, the results reflect implementation in a metropolitan, densely populated area. Additionally, participation was voluntary and thus may have included more highly motivated pharmacists, making it therefore not representative of all pharmacists. Although dropout pharmacists were also interviewed, the opinions of pharmacists who did not participate in the project were not collected. The effectiveness results should be considered with caution. This is due to the very limited number of participants identified as being at high risk who agreed to be contacted and were reachable. Furthermore, the lack of fidelity to eligibility criteria should be addressed to avoid the risk assessment of patients already in medical follow-up. A larger case study should be conducted that takes into consideration preventive strategies and closely evaluates the effectiveness of referrals, attendance and medical outcomes of patients identified as being at high risk. Additionally, the cholesterol value of the patient should be measured with point-of-care testing during the risk assessment. This approach would determine the patient’s SCORE precisely and result in a more reliable cardiovascular risk determination. Finally, further studies should investigate general practitioners’ perceptions of and recommendations on the risk assessment to ensure their close collaboration and thus guarantee successful implementation.

## 5. Conclusions

Considering the increased strain on primary care, pharmacists can take a more active role in identifying, counselling, and referring patients with previously undiagnosed conditions and in guiding at-risk patients to prevent further disease progression. Overall, the patients’ positive experience of the risk assessment and the participating pharmacists’ enthusiasm are encouraging for the sustainability of pharmacist-led preventive services. The community pharmacy was perceived as highly accessible and can lower the threshold for risk evaluation. At the same time, the manageability of performing the service demonstrated its feasibility in daily practice. External support strategies such as interprofessional workshops, adapted software and an electronic data-sharing platform, additional training and broad-based media campaigns should be considered. These could increase awareness of the pharmacists’ new role, service adoption by the staff, and collaboration with general practitioners to ensure the follow-up of patients identified as being at high risk. Financial incentives and remuneration for pharmacists and an extended legal framework for the reimbursement of POCT should be discussed at the political level as they will be critical determinants of the sustainability and larger-scale implementation of the risk-assessment service on a national level. Finally, further studies should evaluate the effectiveness and monitor the implementation of preventive-related services to provide suitable strategies over time to overcome barriers across different implementation stages.

## Figures and Tables

**Figure 1 ijerph-19-08699-f001:**
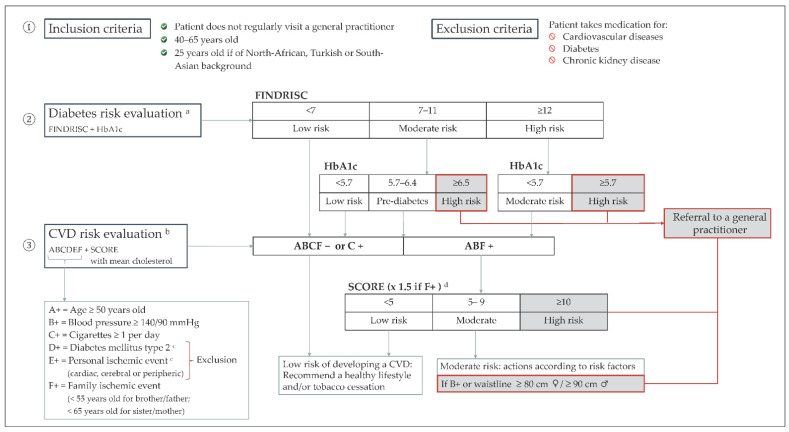
Protocol of the risk-assessment service. Note: ^a^ The diabetes risk evaluation was based on the combination of the FINDRISC questionnaire [20] and a haemoglobin A1c reading. ^b^ The cardiovascular diseases (CVD) evaluation was based on the Boland algorithm [21] and the calibrated Systematic Coronary Risk Evaluation (SCORE) chart for Belgium [22]. ^c^ As patients taking medication for diabetes and cardiovascular diseases were not eligible, the risk factor of diabetes mellitus type 2 [D] and a personal ischaemic event [E] were not integrated into the evaluation of cardiovascular diseases. ^d^ Where there was a familial ischaemic event [F], the score was multiplied by 1.5 (150%), as advised by the recommendations on good clinical practices [23].

**Table 1 ijerph-19-08699-t001:** Description of the collected outcomes classified following the RE-AIM dimensions.

Dimension and Definition	Quantitative Data	Qualitative Data
**Reach** *(The absolute number, proportion, and representativeness of individuals willing to participate in a given intervention)*	- Number of patients screened - Description of the patients’ characteristics - The proportion of the different risk profiles - The proportion of patients identified as high risk	- Barriers and facilitators to patient recruitment - Reasons for participating, according to patients - Reasons for refusal, according to pharmacists
**Effectiveness** *(The impact of an intervention on important outcomes, including potential negative effects, quality of life, and economic outcomes)*	- Outcome of the medical follow-up of the patients identified as high risk	- Patient attitudes and perceptions during the risk assessment
**Adoption** *(The absolute number, proportion, and representativeness of: (a) settings; and (b) intervention agents (people who deliver the programme) who are willing to initiate a programme)*	- Number of participating pharmacies - Number of dropouts during the project ^b^	- Reasons for pharmacists participating in the project - Reasons for dropouts during the project
**Implementation** *(The interventions agents’ fidelity to the various elements of an intervention’s protocol)*	- Adherence to the protocol of the pilot project	- Adherence to the protocol of the pilot project - Internal organisation and adaptation to implement the project - The time needed to provide the service - Facilitators and barriers to implementation
**Likelihood of maintenance ^a^** *(The extent to which: (a) a behaviour is sustained or more after intervention; and (b) a programme becomes institutionalised or part of the routine organisational practices).*	/	- Pharmacists’ experiences with the screening programme - Patients’ acceptance of the service - Sustainability factors

Note: ^a^ Due to the nature of a screening pilot project, the evaluation of the maintenance dimension was adapted to assess the extent to which the project could become institutionalised or part of the routine practice of pharmacists. ^b^ A dropout was defined as a pharmacy that had not recorded any patients by the end of the pilot project.

**Table 2 ijerph-19-08699-t002:** Diabetes risk evaluation—according-to-protocol analysis.

				Diabetes Risk Profile			
				Low	Moderate	High	Undetermined
FINDRISC ^a^	*n (%)*	HbA1C	*n (%)*	*n (%)*	*n (%)*	*n (%)*	*n (%)*
<7	47 *(12.5%)*	/	/	47 *(12.5%)*			
7–11	128 *(34.0%)*	<5.7	98 *(26.1%)*	98 *(26.1%)*			
		5.7–6.4	26 *(6.9%)*		26 *(6.9%)*		
		≥6.5	2 *(0.5%)*			2 *(0.5%)*	
		No data	2 *(0.5%)*				2 *(0.5%)*
≥12	201 (*53.5%)*	<5.7	126 *(33.5%)*		126 *(33.5%)*		
		≥5.7	72 *(19.1%)*			72 *(19.1%)*	
		No data	3 *(0.8%)*				3 *(0.8%)*
	376 *(100%)*		376 (*100%)*	145 *(38.6%)*	152 *(40.4%)*	74 *(19.7%)*	5 *(1.3%)*

Note: ^a^ Finnish Diabetes Risk Score [20].

**Table 3 ijerph-19-08699-t003:** Cardiovascular risk evaluation—according-to-protocol analysis.

			Cardiovascular Risk Profile			
			Low	Moderate	High	Undetermined
		*n (%)*	*n (%)*	*n (%)*	*n (%)*	*n (%)*
**ABCF− or C+ ^a^**	ABCF−	67 *(26.3%)*	67 *(26.3%)*			
**ABF+ ^b^**	Score < 5	152 *(59.6%)*	152 *(59.6%)*			
	Score 5–9	24 *(9.4%)*		24 *(9.4%)*		
	Score ≥ 10	6 *(2.4%)*			6 *(2.35%)*	
	Missing score	6 *(2.4%)*				6 *(2.4%)*
		255 *(100%)*	219 *(85.9%)*	24 *(9.4%)*	6 *(2.4%)*	6 *(2.4%)*

Note: ^a^ Absence of the following risk factors: age [A], blood pressure [B], cigarette smoking [C], familial ischaemic event [F]; or presence of the smoking-related risk exclusively [C+]. ^b^ Presence of one of the following risk factors: age [A], blood pressure [B], familial ischaemic event [F].

**Table 4 ijerph-19-08699-t004:** Effectiveness—according-to-protocol analysis.

51	Patients with High Level of Risk for Diabetes or Cardiovascular Diseases	
	18	Did not consent to be contacted by the research team
**33**	**Possible inclusions**	
13 lost to Follow-up	3	No contact details
	2	Wrong contact details
	8	Unsuccessful contact attempts
**20**	**Inclusions**	
	2	Not aware that there was a high risk
	3	Had not (yet) gone to a general practitioner, but aware that there was a high risk
	3	Went to a general practitioner for another reason; high risk not discussed as not considered a priority
	6	Already in medical follow-up for cardiovascular disease/risk of diabetes
	5	Went to a general practitioner/specialist, with a negative outcome on diagnosis
	1	Diagnosis

## Data Availability

Not applicable.

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
