# Peer review of "Diabetes and Cardiovascular Diseases Risk Assessment in Community Pharmacies: An Implementation Study"

_ijerph, 2022, doi:10.3390/ijerph19148699_

Round 1
Reviewer 1 Report
This is an article with the nice potential, however, prior publication there are some issues that should be addressed.
INTRODUCTION
Line 36: instead of ,,are slow progression’’ should be ,,have slow progression’’ or ,,are slow progressing’’
METHODS
Generally, this is the hardest section to follow. Try to reorganize it. Start with the description of the exact study period and the study components. Then describe each of the component separately.
Some minor issues in this section:
,,During the six months’’, please add which six months.
How many focus groups were conducted in total?
RESULTS
A combination of the FINDRISC questionnaire and HbA1c measure 248 was used to determine the patients’ risk of developing diabetes within the next 10 years- this is the description of the instrument used and should be in the methods section.
Author Response
Dear reviewer,
We appreciate the time and effort you and others reviewers dedicated to providing feedback on our manuscript and are grateful for the insightful comments and valuable improvements to our paper.
Please find in the attachment our point-by-point responses to your comments and concerns. All page numbers refer to the revised manuscript file with tracked changes.

Reviewer 2 Report
This submission by Rondeaux, et al provides an outstanding approach to improve early detection of diabetes and in turn cardiovascular risk in a urban and suburban area of a large city using evaluation and counseling by community pharmacists. A large number of pharmacies participated (37) and 502 patients were included. This means about 10 pts for each pharmacy.
Two points should be emphasized; 1 the selection criteria were adapted to socially disadvantaged people; 2 such a screening program is intended to be appointment-free. However it has another advantage, making the pharmacists'work more relevant and, accordingly more appealing. This sould probably be added in the discussion section. As a physician, e.g., I allways appreciate when pharmacists are calling to ask a question on my prescription (or suggest to change it..). This team working arrangement is obviously a quality criterion. Giving an opportunity to improve early detection of T2DM and CVRF to pharmacists is a major step in management of otherwise less medicalized pts.
It may be useful for nonspecialized readers to enlist the FINDRISC score items. Indeed, the submission is a very long manuscript whereas screening T2DM in a pharmacy should be suitable for most pharmacists or event their assitants. They should understand, while reading the paper that this screening is not too sophisticated, and easy to perform. This point should probably be added to the abstract.
Author Response

(The authors gave the same response as above.)

Reviewer 3 Report
The authors have conducted a study to present a new role of pharmacist in assessing risk of CVD and diabetes in community pharmacies. The idea is novel and the concept appears to be promising based on the outcomes presented in this manuscript. I congratulate the authors for such an excellent piece of work.
Strengths: novel idea and provides solution to the critical need for screening chronic diseases like T2DM and CVD at early stages. Patients visit pharmacists more often than any other healthcare provider. Therefore, screening at the level of community pharmacists is a smart move. Even though the idea here does not provide any complicated system to deal with, this role opens door for a specialty pharmacist in the future.
Weaknesses: the manuscript is too long and have several instances of grammatical errors. I would recommend the authors to utilize English service to improve the readability of the manuscript. I cannot keep listing all the errors but some of the examples are listed below so that authors can fix them all over the manuscript.
Ln 55 – “location” should be “locations”; add comma after hours
Ln 68 – “environment should be “environments”
Lns 86-88 – aim should be reworded for clarity. Change “a diabetes” to “diabetes”
Ln 106 – use “the patient risk profile” or “patient’s risk profile”
Ln 120 – change “into the cardiovascular diseases evaluation” to “in the evaluation of the cardiovascular disease”
Lns 216-217 – add comma after day. i.e., January 1st, 2022 – use this format
Ln 293 – change “pharmacist” to “pharmacists”
Ln 331 – “with five who weren’t diagnose with diabetes” – this is not professional language. Use proper language. “with five who were not diagnosed with diabetes”
Ln 366 – change “afterwards” to “afterward”
Ln 423 – change “deepen” to “deepening”
Ln 424 – change “the pharmacist” to “pharmacist”
Ln 433 – change “service” to “services”
Ln 437 – same
Ln 473 – change “the pharmacist” to “pharmacist”
Ln 478 – change “screening” to “screenings”
Author Response
Dear reviewer,
We appreciate the time and effort you and other reviewers dedicated to providing feedback on our manuscript and are grateful for the insightful comments and valuable improvements to our paper.
Please find in the attachment our point-by-point responses to your comments and concerns. All page numbers refer to the revised manuscript file with tracked changes.

Reviewer 4 Report
Primary prevention of cardiovascular disease in Europe is largely lacking. One of the possible causes of this phenomenon is the difficulty for patients to quickly access a risk stratification program. In this descriptive paper (statistical analysis is almost absent) authors report their experience on the involvement of pharmacies in a screening program.
The study case series is rather limited, many patients were lost to follow-up, and, from the data shown, screening likely changed the prognosis in only one case allowing diagnosis of unknown diabetes. All in all, a modest result. In this light, this study should be presented as a feasibility study; the real effectiveness in changing the outcome of screened patients will have to be evaluated in a subsequent study with larger case series. A datum that does not seem to have been explored is the reception of general practitioners to this initiative; it is difficult to imagine the success of such a program without their involvement and collaboration.
Overall this paper is too long with many repetitions and should be shortened by at least 20-25%
Specific comments
Lines 132-4. The authors state that “Since the cholesterol value was not measured during this project, the mean of the Belgian cholesterol value for men and women was used to evaluate the score of the patients”. This attribution of cholesterol level is completely arbitrary and makes the calculation of SCORE risk less reliable. Why wasn't total cholesterol assayed using a point-of-care system? In my view, the lack of such data seriously invalidates the calculation of cardiovascular risk and consequently all provisions that are guided by such risk assessment. This limitation should be made explicit and remembered in the final discussion.
Line 151. The sentence “the dialogue between the pharmacist and their patients” is incorrect. Pharmacists do not have their own patients. The correct sentence should be “the dialogue between the pharmacist and patients”. The same correction to line 355
Line 334. Table 4. The number 18 is not spelled correctly (i.e., the numbers 1 and 8 are overlapping instead of side by side)
Author Response

(The authors gave the same response as above.)

Reviewer 5 Report
Thank you for allowing me to review this manuscript. This manuscript entitled "Assessment of the risk of diabetes and cardiovascular diseases in community pharmacies: an implementation study". This article aims to present the results of the implementation study of a pilot project for a diabetes and cardiovascular disease risk assessment service in Belgian community pharmacies.
It is an interesting and highly relevant article today, although it has several limitations that make it suitable for publication in this journal. These limitations are detailed below:
- I would recommend that the authors justify the novelty and relevance of the study being carried out in more detail in the Introduction
- It would be interesting to include in more detail information on the local context for data collection. Why is it advisable to collect data there? What are the reasons for doing it?
- In the material and methods section, it would be interesting to provide more information on the questionnaires used for data collection. I would recommend the authors to specify the psychometric properties of such questionnaires. Also, it would be important to point out the inclusion and application requirements among the participants. On the other hand, I would recommend justifying the sample size in more detail.
- In the results section, I would recommend putting a table footer in the tables, where the acronyms of the tables will be specified.
- The conclusions are very elaborated, they are clear and precise. It would be interesting to further highlight the importance and impact of the study topic in the clinical setting. Also, I would recommend the authors to include a proposal for the future to continue in this field of research.
Author Response

(The authors gave the same response as above.)

Round 2
Reviewer 1 Report
Authors have responded to the most of my requests.
I would only suggest to add both quantitative and qualitative part of the study to the description at the beginning of the methods section. This way the qualitative part kind of surprises the readers.
Author Response
Dear reviewer,
Thank you for re-reading our manuscript and for your suggestion.
Accordingly, we added a new sentence at the very beginning of the method section. This addition (highlighted) can be seen in line 103 as follows:
"From October 2020 to February 2021, a selection of Brussels pharmacists invited patients to participate in opportunistic screening for diabetes and cardiovascular disease risk factors. They specified this new service was part of a qualitative and quantitative research study to evaluate its feasibility in community pharmacies. The screening followed a multi-step approach..."
Hopefully, this introduces briefly the method used at the beginning, while the precisions of the study design are presented in its own section a bit later on line 177.
Reviewer 4 Report
The revised paper is shorter and more readable than the first draft.
The many limitations of this study remain unchanged, which, moreover, are acknowledged and discussed extensively by the authors, who have better specified how this is a feasibility study that therefore does not allow definitive conclusions about the effectiveness of this health strategy.
In the previous review I wrote: “The sentence “the dialogue between the pharmacist and their patients” is incorrect. Pharmacists do not have their own patients. The correct sentence should be “the dialogue between the pharmacist and patients”. The same correction to line 355. Authors replay that “we do believe that pharmacists perceive patients coming regularly as their patients in the same way as doctors do”. I equally respectfully stand by my opinion emphasizing that words are important to avoid confusion of roles. In the Oxford dictionary, patient is defined as “a person who receives treatment from a particular doctor, dentist, etc.” The pharmacist's role is not to make diagnoses and prescribe therapies. They certainly make a significant contribution in caring for patients, but strictly speaking they are not "their" patients. I personally (and respectfully) consider this an important and not secondary clarification and I think the text should be corrected accordingly
Specific comments.
Line 64. The sentence 错误!è¶…é“¾æŽ¥å¼•ç”¨æ— æ•ˆ written in ideograms is a typo
Author Response
Dear reviewer,
Thank you for re-reading our manuscript.
As we do not wish any confusion about the role of the pharmacists, we corrected all sentences accordingly to your suggestion. A word search has been conducted to detect every occurrence. As such, the lines have been changed as follows :
Line 101: "... pharmacists invited patients to participate ..." instead of "... pharmacists invited their patients to..."
Line 169: "... motivational techniques for patient counselling." instead of "...techniques to motivate their patients."
Line 308: "... in proposing the programme to patients..." instead of "... in proposing the programme to their patients..."
Line 355: "... as an opportunity to show their added-value." instead of "...as an opportunity to show their added-value to their patients."
Line 403: "... the risk assessment allowed the pharmacists to spend time with the patients..." instead of "...the risk assessment allowed the pharmacists to spend time with their patients..."
Line 447: "... to overstep their role in the eyes of the patients..." instead of "... to overstep their role in the eyes of their patients..."
Concerning line 64, it seems the issue with the ideograms was coming from a faulty hyperlink from the reference citation manager. We corrected the problem by deleting the citation and inserting it again.